# 4-Band Multispectral Images Demosaicking Combining LMMSE and Adaptive Kernel Regression Methods

**DOI:** 10.3390/jimaging8110295

**Published:** 2022-10-25

**Authors:** Norbert Hounsou, Amadou T. Sanda Mahama, Pierre Gouton

**Affiliations:** 1Institute of Mathematics and Physical Sciences, University of Abomey-Calavi, Porto-Novo BP 613, Benin; 2Science and Technology Faculty, University of Burgundy, 21078 Dijon, France

**Keywords:** demosaicking algorithm, multispectral filter array, LMMSE, adaptive kernel regression

## Abstract

In recent years, multispectral imaging systems are considerably expanding with a variety of multispectral demosaicking algorithms. The most crucial task is setting up an optimal multispectral demosaicking algorithm in order to reconstruct the image with less error from the raw image of a single sensor. In this paper, we presented a four-band multispectral filter array (MSFA) with the dominant blue band and a multispectral demosaicking algorithm that combines the linear minimum mean square error (LMMSE) and the adaptive kernel regression methods. To estimate the missing blue bands, we used the LMMSE algorithm and for the other spectral bands, the directional gradient method, which relies on the estimated blue bands. The adaptive kernel regression is then applied to each spectral band for their update without persistent artifacts. The experiment results demonstrate that our proposed method outperforms other existing approaches both visually and quantitatively in terms of peak signal-to-noise-ratio (PSNR), structural similarity index (SSIM) and root mean square error (RMSE).

## 1. Introduction

Digital color cameras generally sensitive to three bands of the visible electromagnetic spectrum are used to capture digital color images representing the reflectance of the observed object. Nowadays, technological advancement has made it possible to overcome this three-band limitation with the development of multispectral digital cameras to acquire multispectral images with more than three spectral bands per pixel. There are several types of multispectral image acquisition systems including single-sensor one-shot cameras which are equipped with a multispectral filter mosaic. However, in the raw image from the sensor, each pixel is characterized by a single available spectral band. We will have to reconstruct the missing spectral bands by the demosaicking method. The reconstruction performance depends on the optimal choice of MSFA and the multispectral demosaicking algorithm.

Several MSFA patterns are proposed in the literature. To our knowledge, Miao et al. [1] are the first to propose a generic MSFA model from a binary tree by recursively separating the checkerboard pattern based on a tree decomposition which defines the number of spectral bands and the probability of occurrence of each band. Aggarwal et al. [2] meanwhile implemented two MSFA patterns, one random and the other uniform, which can be generalized to any number of bands. In [3], Monno et al. proposed a five-band MSFA based on the dominant G band requirement, which is used by Jaiswal et al. in their multispectral demosaicking algorithm [4]. To overcome the difficulties in combining spectral resolution and spatial correlation, Mihoubi et al. proposed a 16-band MSFA without a dominant spectral band [5]. Recently, Bangyong et al. designed a uniform four-band MSFA pattern [6] with the same probability of occurrence for each band and a nine-band MSFA pattern [7] in which one band is dominant and the other eight have the same probability of occurrence arranged in a 4 × 4 mosaic.

Many multispectral demosaicking algorithms using the designed MSFAs have been implemented in the literature [8]. Miao et al. [9] proposed a binary tree-based edge-sensing (BTES) multispectral demosaicking algorithm that recursively performs binary tree-based edge detection interpolation. However, the performance of this algorithm in classical edge detection interpolation is limited. Recently, Monno et al. [10,11,12] proposed a series of demosaicking algorithms for its proposed five-band MSFA. The first of these algorithms [10] developed several guide images that were used in the interpolation of the different spectral bands. The authors used residual interpolation to generate a guide image for structure-preserving interpolation [11] and proposed adaptive residual interpolation by adaptively combining two algorithms based on residual interpolation and selecting an appropriate number of iterations for each pixel [12]. Jaiswal et al. [4] used the high-frequency component of the G-band to interpolate the other bands based on an inter-band correlation analysis while Mihoubi et al. [2] proposed a 16-band MSFA algorithm based on a pseudo-panchromatic image (PPI), which is estimated by applying an averaging filter to the raw image and then adjusted such that the PPI values are correlated. The difference between each available value of the adjusted raw image and PPI is calculated. The calculated local directional weights are then used to estimate the fully defined difference using an adaptive weighted bilinear interpolation. Each band is finally estimated by adding a PPI and the difference. In [6], a method of applying directional interpolation along the edges of an image was proposed. In this method, the image edges are calculated from the raw image to define the direction interpolation with the neighbors. Considering the features of the filter arrays, image edges, and a constant hue, the missing bands per pixel were recovered from the existing bands. Then, the image is separated into high-and low-frequency components by applying a wavelet transform, and the high-frequency images that are highly correlated are modified using luminance information to refine the demosaicked image. In [7], a multispectral algorithm that estimates the missing dominant band at each spatial position with a weighted average of the neighboring values of the dominant band was described. The dominant band reconstructed at different spatial positions is then used as a guided image to estimate all other missing bands using the guided filter and residual interpolation.

Multispectral images demosaicked using the previous algorithms suffer from severe artefacts in edge regions. To overcome these limitations, a new avenue of multispectral demosaicking called the LMMSE method is being explored. Zhang and Wu [13], in the demosaicking of color images such as Bayer’s CFA [14], had developed the LMMSE method which is based on the assumption that the gradient of the G and R/B channels correspond to low-pass filtering, given their strong correlation. The LMMSE adaptively estimates the missing G values in both horizontal and vertical directions and then merges them optimally. A very interesting result is the introduction of the neighborhood in the LMMSE formulation by Amba et al. [15] for color images they recently extended to eight-band multispectral demosaicking by applying a linear operator that minimizes the root mean square error between the reconstructed image and the original raw image [16]. This linear operator multiplied by the MSFA image provides an estimate of the reconstructed image. According to [16], the LMMSE method constitutes a good potential candidate for real-time applications because, after training, it could be integrated into the equipment of the camera and operate in real time without losing the generality required by the various provisions present on the market.

The contributions of our paper focused on the LMMSE method in [13] and the adaptive kernel regression kernel as described in [17] are of three kinds. (1) We identified with justification a generic four-band MSFA with the dominant blue band for our multispectral demosaicking algorithm. (2) We proposed the directional LMMSE method for estimating the missing blue bands and the directional gradient method for the other three spectral bands. (3) To take into account the details at the edges and the denoising of the reconstructed image, we have successfully combined the LMMSE method with the adaptive kernel regression. This paper is organized as follows: in the second section, we justify our proposed four-band MSFA and the application assumptions of the LMMSE method. The existing LMMSE method and the adaptive kernel regression used are described in the third section. The proposed algorithm and the experimental results are, respectively, presented in the fourth and fifth sections.

## 2. Design of the Four-Band MSFA and Application Assumptions of the LMMSE Method

### 2.1. Design of the 4-Band MSFA

The identified four-band MSFA is based on the generic method of Miao et al. [1] based on the binary tree with the probability of occurrence of each spectral band (Figure 1). The multispectral images of the cave dataset [18] used in our simulations are acquired with a camera whose sensor is fitted with the liquid crystal tunable filter (LCTF) [19] (Figure 2) such as the energy of the blue band of wavelength λ = 450 nm is very weak compared to the other bands, followed by the orange band whose wavelength is λ = 600 nm. The red band (λ = 700 nm) has the greatest energy preceded by the green band (λ = 550 nm). According to [20], the energy imbalance between the different spectral bands produces in the demosaicked image, severe degradation of the low-energy bands due to their sensitivity to noise. It then appears necessary that a balancing be carried out to optimize the shape of the transmittance filters. To balance the energies of the different spectral bands and avoid degradation of the low energy bands, we opted for an MSFA with a dominant blue band with a probability of occurrence of 1/2 followed by the orange band with a probability of occurrence of 1/4, and 1/8 is the probability of occurrence of the red and green bands.

After balancing the energies of the different spectral bands according to the proposed MSFA pattern, their spectral sensitivities are shown in Figure 3.

### 2.2. Application Assumptions of LMMSE Method

The LMMSE method as used for RGB images obeys two assumptions. Firstly, in natural images, the different spectral bands are strongly correlated. Then, the gradient of the different bands remains constant and constitutes a smooth process (low pass) [13]. To verify these assumptions, we first determined in Table 1, the spectral correlation of the different spectral bands of the cave dataset [18] multispectral images used. The spectral correlation of two bands is best if the correlation coefficient between these two bands is between 0.5 and 1. From the analysis of Table 1 and with a few minimal exceptions, all the spectral bands of the different multispectral images are strongly correlated. Secondly, we have plotted the power spectral functions of the gradients of the different bands for three multispectral images (Figure 4, Figure 5 and Figure 6). When we analyze these different spectral functions, we have realized that the power of the different gradient signals is concentrated in the low-frequency band then that each of these functions has a peak around the zero frequency. Thus, the two above-mentioned are therefore verified for our simulation multispectral images.

## 3. Overview of LMMSE and Kernel Regression Methods

### 3.1. LMMSE Demosaicking Method

In their color image demosaicking algorithm, Zhang and Wu [13] used the LMMSE method to estimate the missing *G* bands at different pixels. We briefly present here the estimate of the *G*-band at each spatial position (i,j) of the red pixels in the CFA image. Thus, the missing *G*-band at the red pixels is obtained according to the formula used:(1)$${\hat{G}}_{i,j}=R_{i,j}+{\hat{\Delta }}_{g,r}\left(i,j\right)$$

The gradient $\Delta_{g,r}$ of the red and green bands is estimated in both horizontal and vertical directions such as:(2)$${\hat{\Delta }}_{g,r}{}^{h}\left(i,j\right)=\{\begin{array}{c} {\hat{G}}_{i,j}{}^{h}-R_{i,j}{}^{h},\;\;if\;G\;is\;interpoleted\;\\ G_{i,j}{}^{h}-{\hat{R}}_{i,j}{}^{h},\;\;\;if\;R\;is\;interpoleted \end{array}$$
(3)$${\hat{\Delta }}_{g,r}{}^{v}\left(i,j\right)=\{\begin{array}{c} {\hat{G}}_{i,j}{}^{v}-R_{i,j}{}^{v},\;\;if\;G\;is\;interpoleted\;\\ G_{i,j}{}^{v}-{\hat{R}}_{i,j}{}^{v},\;\;if\;R\;is\;interpoleted \end{array}$$

Before calculating the gradient, a second order Laplacian interpolation is used beforehand to know at each pixel, the “missing samples”. The noises associated with the directional gradient estimate are determined as:(4)$$\{\begin{array}{c} \varepsilon_{g,r}{}^{h}\left(i,j\right)=\Delta_{g,r}\left(i,j\right)-{\hat{\Delta }}_{g,r}{}^{h}\left(i,j\right)\\ \varepsilon_{g,r}{}^{v}\left(i,j\right)=\Delta_{g,r}\left(i,j\right)-{\hat{\Delta }}_{g,r}{}^{v}\left(i,j\right) \end{array}$$

So, we have:(5)$$\{\begin{array}{c} {\hat{\Delta }}_{g,r}{}^{h}\left(i,j\right)=\Delta_{g,r}\left(i,j\right)-\varepsilon_{g,r}{}^{h}\left(i,j\right)\\ {\hat{\Delta }}_{g,r}{}^{v}\left(i,j\right)=\Delta_{g,r}\left(i,j\right)-\varepsilon_{g,r}{}^{v}\left(i,j\right) \end{array}$$

The gradient $\Delta_{g,r}$ is estimated by the LMMSE method. Let us denote by x the gradient $\Delta_{g,r}$, y the associates ${\hat{\Delta }}_{g,r}{}^{h}$ and ${\hat{\Delta }}_{g,r}{}^{v}$ and ϑ the associated noises $\varepsilon_{g,r}{}^{h}$ and $\varepsilon_{g,r}{}^{v}$. The Equation (5) becomes:(6)y(i,j)=x(i,j)+ϑ(i,j)

The optimal estimate of the minimum mean square error (MMSE) of *x* is defined:(7)$$\hat{x}=E\left[x/y\right]=\int^{\;}xp\left(x/y\right)dx$$

However, in practice, the probability *p*(*x*/*y*) is rarely known making estimation of MMSE difficult. Therefore, instead of MMSE, the authors used the LMMSE method to estimate *x*, such as:(8)$$\hat{x}=E\left[x\right]+\frac{Cov\left(x,y\right)}{Var\left(y\right)}\left(y-E\left[y\right]\right)$$

E[x] is the mathematical expectation of x, Cov(x,y) the covariance and Var(y) the variance of y. By setting $\mu_{x}=E\left[x\right]$, $\sigma_{x}{}^{2}=Var\left(x\right)$ and $\sigma_{\vartheta }{}^{2}=Var\left(\vartheta \right)$, the Equation (8) becomes:(9)$$\hat{x}=\mu_{x}+\frac{\sigma_{x}{}^{2}}{\sigma_{x}{}^{2}+\sigma_{\vartheta }{}^{2}}\left(y-\mu_{x}\right)$$

For an optimal estimate of $\hat{x}$, the latter is estimated adaptively by merging the values determined in both horizontal and vertical directions in the neighborhood of *y*. Denoting by ${\hat{x}}_{h}\left(n\right)$ and ${\hat{x}}_{v}\left(n\right)$ the horizontal and vertical LMMSE estimates of x obtained from the Equation (9), then by $w_{h}$ and $w_{v}$ the both horizontal and vertical weights, respectively, the optimal LMMSE estimate of x is defined by:(10)$${\hat{x}}_{w}\left(i,j\right)=w_{h}\left(i,j\right).{\hat{x}}_{h}\left(i,j\right)+w_{v}\left(i,j\right).{\hat{x}}_{v}\left(i,j\right)$$

With $w_{h}\left(i,j\right)+w_{v}\left(i,j\right)=1$ to minimize the estimation error.
(11)$$\{\begin{array}{c} \;w_{h}\left(i,j\right)=\frac{\sigma_{{\tilde{x}}_{v}}{}^{2}\left(i,j\right)}{\sigma_{{\tilde{x}}_{h}}{}^{2}\left(i,j\right)+\sigma_{{\tilde{x}}_{v}}{}^{2}\left(i,j\right)}\\ w_{v}\left(i,j\right)=\frac{\sigma_{{\tilde{x}}_{h}}{}^{2}\left(i,j\right)}{\sigma_{{\tilde{x}}_{h}}{}^{2}\left(i,j\right)+\sigma_{{\tilde{x}}_{v}}{}^{2}\left(i,j\right)} \end{array}$$

${\tilde{x}}_{h}$ and ${\tilde{x}}_{v}$ are the estimation errors of ${\hat{x}}_{h}$ and ${\hat{x}}_{v}$ such that:(12)$$\{\begin{array}{c} {\hat{x}}_{h}\left(i,j\right)=x\left(i,j\right)-{\tilde{x}}_{h}\left(i,j\right)\\ {\hat{x}}_{v}\left(i,j\right)=x\left(i,j\right)-{\tilde{x}}_{v}\left(i,j\right) \end{array}$$

$\sigma_{{\tilde{x}}_{h}}{}^{2}$ and $\sigma_{{\tilde{x}}_{v}}{}^{2}$ are, respectively, the variances of ${\tilde{x}}_{h}$ and ${\tilde{x}}_{v}$.

More information is given in [13].

### 3.2. Kernel Regression Method

Takeda et al. [17] proposed a kernel regression that is used in the iterative reconstruction of color images, and which takes into account limitations such as strong denoising along the edges, the high retention of detail in the edges, and the limited presence of blur in the reconstructed image. The estimate of y pixel at $x_{i}$ location is defined as:(13)$$y_{i}=z\left(x_{i}\right)+\in_{i};\;i=1,2,\ldots \ldots \ldots \ldots ,p$$

$\in_{i}$ is the associated noise and z(.) the regression function obtained by Taylor expansion of N-order.
(14)$$z\left(x_{i}\right)=\beta_{o}+\beta_{1}{}^{T}\left(x_{i}-x\right)+\beta_{2}{}^{T}vech\left\{\left(x_{i}-x\right){\left(x_{i}-x\right)}^{T}\right\}+\ldots$$

vech(.) is a half-vectorization operator of the lower triangular portion of a symmetric matrix such as:(15)$$\{\begin{array}{c} \;\;\;vech\left(\left[\begin{array}{cc} a & b\\ b & d \end{array}\right]\right)={\left[\begin{array}{ccc} a & b & d \end{array}\right]}^{T}\;\;\\ \;vech\left(\left[\begin{array}{ccc} a & b & c\\ b & e & f\\ c & f & i \end{array}\right]\right)={\left[a\;\;b\;\;c\;\;e\;\;f\;\;i\right]}^{T} \end{array}$$

The $\beta_{n}$ are obtained as below:(16)$$\{\begin{array}{c} \beta_{o}=z\left(x\right)\;\;\;\\ \;\;\;\;\;\;\;\;\;\;\;\;\;\;\beta_{1}={\left[\frac{\partial z\left(x\right)}{\partial x_{1}},\frac{\partial z\left(x\right)}{\partial x_{2}}\right]}^{T}\\ \;\beta_{2}=\frac{1}{2}{\left[\frac{\partial^{2}z\left(x\right)}{\partial^{2}x_{1}},2\frac{\partial^{2}z\left(x\right)}{\partial x_{1}\partial x_{2}},\;\frac{\partial^{2}z\left(x\right)}{\partial^{2}x_{2}}\right]}^{T} \end{array}$$

They are computed by the following optimization problem:(17)$$\underset{\left\{\beta_{n}\right\}}{\min}\overset{p}{\underset{i=1}{\sum }}{\left[y_{i}-\beta_{o}-\beta_{1}{}^{T}\left(x_{i}-x\right)-\beta_{2}{}^{T}vech\left\{\left(x_{i}-x\right){\left(x_{i}-x\right)}^{T}\right\}-\ldots \right]}^{2}K_{H}\left(x_{i}-x\right)$$
(18)$$K_{H}\left(x_{i}\right)=\frac{1}{det\left(H\right)}K\left(H^{-1}x_{i}\right)$$

*K* is the kernel function and *H* the 2 × 2 smoothing matrix of order defined by:(19)$$H_{i}=h\mu_{i}I$$

h is a global smoothing parameter, $\mu_{i}$ a local density parameter which controls the kernel size and I an identity matrix.

### 3.3. Adaptive Kernel Regression

The adaptive kernel regression is an extension of the classical kernel regression [17] and structured in the same way as in Equation (17) where the classical kernel is replaced by the adaptive kernel.
(20)$$K_{adapt}\left(x_{i}-x\right)\left(y_{i}-y\right)=K_{H_{s}}\left(x_{i}-x\right)K_{h}\left(y_{i}-y\right)$$

$H_{s}=h_{s}I$ is the spatial smoothing matrix. To avoid computational complexity, the order estimation is limited to N = 0. The necessary calculations are then limited to those which estimate the parameter $\beta_{o}$ such as:(21)$$\hat{z}\left(x\right)={\hat{\beta }}_{0}$$

The value of a spectral band at a spatial position is determined by:(22)$$\hat{z}\left(x\right)=\frac{\sum_{i=1}^{P}K_{H_{s}}\left(x_{i}-x\right)K_{h}\left(y_{i}-y\right)y_{i}}{\sum_{i=1}^{P}K_{H_{s}}\left(x_{i}-x\right)K_{h}\left(y_{i}-y\right)}$$

Expressing $K_{adapt}$ in spatial and radiometric terms weakens the performance of the estimate. Consequently, the adaptive kernel is replaced by an adaptive steering kernel, the denoising of which takes place most strongly along the edges.
(23)$$K_{adapt}\left(x_{i}-x,y_{i}-y\right)=K_{H_{i}{}^{steer}}\left(x_{i}-x\right)$$

The steering matrix is defined as:(24)$$H_{i}{}^{steer}=h\mu_{i}C_{i}{}^{-\frac{1}{2}}$$

The $C_{i}$ are symmetric covariance matrix used to temper the blurring effect around edges and whose values are obtained by a differentiation between the value of the central pixel and those of the neighboring pixels. The global smoothing parameter *h* makes it possible to have a strong denoising effect and the steering kernel is a Gaussian kernel.
(25)$$K_{H_{i}{}^{steer}}\left(x_{i}-x\right)=\frac{\sqrt{\det\left(C_{i}\right)}}{2\pi h_{i}{}^{2}\mu_{i}{}^{2}}exp\left\{-\frac{{\left(x_{i}-x\right)}^{T}C_{i}\left(x_{i}-x\right)}{2h_{i}{}^{2}\mu_{i}{}^{2}}\right\}$$
(26)$$C_{i}\approx \left[\begin{array}{cc} \sum_{x_{j}\in w_{i}}z_{x_{1}}\left(x_{j}\right)z_{x_{1}}\left(x_{j}\right) & \sum_{x_{j}\in w_{i}}z_{x_{1}}\left(x_{j}\right)z_{x_{2}}\left(x_{j}\right)\\ \sum_{x_{j}\in w_{i}}z_{x_{1}}\left(x_{j}\right)z_{x_{2}}\left(x_{j}\right) & \sum_{x_{j}\in w_{i}}z_{x_{2}}\left(x_{j}\right)z_{x_{2}}\left(x_{j}\right) \end{array}\right]$$
where $z_{x_{1}}\left(\cdot \right)$ and $z_{x_{2}}\left(\cdot \right)$ are the first derivatives along $x_{1}$ and $x_{2}$ directions and $w_{i}$ is a local analysis window around the position of interest. We set the smoothing parameter $h_{i}$ to 2 to have a strong denoising effect along edges and the local density parameter $\mu_{i}$ to 1 for kernel size control.

## 4. Proposed Multispectral Demosaicking Method

The proposed algorithm is subdivided into six main steps. The blue band being the dominant band of our MSFA (Figure 1c), this band is the first one estimated at the other pixels.

### 4.1. Blue Band Estimation by LMMSE Method

We estimate the blue band missing at the orange pixel by applying the formula:(27)$$\hat{B}\left(i,j\right)=O\left(i,j\right)+{\hat{\Delta }}_{b,o}\left(i,j\right)$$

The gradient $\Delta_{b,o}$ is interpolated by the LMMSE method of Equations (10) and (11). We adopt the same strategy to estimate blue bands at red and green pixels such as:(28)$$\hat{B}\left(i,j\right)=\{\begin{array}{c} R\left(i,j\right)+{\hat{\Delta }}_{b,r}\left(i,j\right),\;\;\;\;\;\;\;\;\;\;in\;R\;pixels\;\\ G\left(i,j\right)+{\hat{\Delta }}_{b,g}\left(i,j\right),\;\;\;\;\;\;\;\;\;in\;G\;pixels \end{array}$$

### 4.2. Orange Band Estimation at Red and Green Pixels

The green and red bands have identical neighborhoods.

The gradient values $\Delta_{b,o}$ in the four directions (Figure 7) northwest (nw), northeast (ne), southwest (*sw*) and southeast (se)  in the neighborhood of a green or red pixel are, respectively, denoted by:(29)$${\hat{\Delta }}_{bo}\left(i,j\right)=\frac{\Delta^{nw}{}_{bo}\left(i,j\right)+\Delta^{ne}{}_{bo}\left(i,j\right)+\Delta^{sw}{}_{bo}\left(i,j\right)+\Delta^{se}{}_{bo}\left(i,j\right)}{4}$$

Each orange band is then estimated at the green and red pixels by the formula:(30)$$\hat{O}\left(i,j\right)=\hat{B}\left(i,j\right)-{\hat{\Delta }}_{bo}\left(i,j\right)$$

### 4.3. Green Band Estimation at Red Pixels and Vice Versa

We apply the same strategies as before but in a wider neighborhood (Figure 8) in the north (*n*), south (*s*), east (*e*), and west (*w*) directions.
(31)$$\hat{R}\left(i,j\right)=\hat{B}\left(i,j\right)-{\hat{\Delta }}_{br}\left(i,j\right)$$
(32)$$\hat{G}\left(i,j\right)=\hat{B}\left(i,j\right)-{\hat{\Delta }}_{bg}\left(i,j\right)$$

### 4.4. Red and Green Bands Estimation at Orange Pixels

The Figure 9 shows the neighborhood of orange band for original and estimated red and green pixels.

Taking into account the neighborhood of the orange bands for the red and green pixels (Figure 9), original or estimated, we estimate the red and green spectral bands for the orange pixels according to the formulas:(33)$$\hat{R}\left(i,j\right)=\hat{B}\left(i,j\right)-{{\hat{\Delta }}^{\prime }}_{br}\left(i,j\right)$$
(34)$$\hat{G}\left(i,j\right)=\hat{B}\left(i,j\right)-{{\hat{\Delta }}^{\prime }}_{bg}\left(i,j\right)$$
where ${{\hat{\Delta }}^{\prime }}_{ij}$ and ${{\hat{\Delta }}^{\prime }}_{bg}$ are respectively the gradient values in the four directions (Figure 9b,c) for the red and green pixels.

### 4.5. Red, Green, and Orange Bands Estimation at Blue Pixels

From Figure 10b–d, we can see a symmetry of the neighborhood of the blue pixels for the red, green and orange pixels. If we denote by $\Delta^{n}{}_{bo}\;$; $\;\Delta^{s}{}_{bo}$; $\Delta^{w}{}_{bo}$ and $\Delta^{e}{}_{bo}$ the directional gradients of the blue and orange pixels in a neighborhood, we compute the average of the gradient bilinearly as follows:(35)$${\hat{\Delta }}_{bo}\left(i,j\right)=\frac{\Delta^{n}{}_{bo}\left(i,j\right)+\Delta^{s}{}_{bo}\left(i,j\right)+\Delta^{e}{}_{bo}\left(i,j\right)+\Delta^{w}{}_{bo}\left(i,j\right)}{4}$$

Therefore, the missing orange band at the blue pixels is estimated by the relation:(36)$$\hat{O}\left(i,j\right)=B\left(i,j\right)-{\hat{\Delta }}_{bo}\left(i,j\right)$$

Similarly, we estimate the red and green bands at the blue pixels such as:(37)$$\hat{R}\left(i,j\right)=B\left(i,j\right)-{\hat{\Delta }}_{br}\left(i,j\right)$$
(38)$$\hat{G}\left(i,j\right)=B\left(i,j\right)-{\hat{\Delta }}_{bg}\left(i,j\right)$$

### 4.6. Estimated Bands Enhancement Using Adaptive Kernel Regression

Although the previous estimation formulas have worked well for color images as in [13], their use for multispectral images remains limited, especially not taking into account details in the strong edge or rich texture. To correct these imperfections, each estimated spectral band is refined by using the adaptive kernel where we replaced the refined Equation (21) with the formula as defined in [21]. Thus, the refined a spectral band at $x_{p}$ location is defined by:(39)$$\hat{z}\left(x_{p}\right)=\frac{1}{w_{x_{p}}}\underset{x_{i}\epsilon N_{p}}{\sum }K_{H_{i}{}^{steer}}\left(x_{i}-x_{p}\right)M\left(x_{i}\right)S\left(x_{i}\right)$$
where $N_{p}$ is the set of neighbor pixel locations of the location $x_{p}$, $S\left(x_{i}\right)$ the sampled value at the location $x_{i}$, $M\left(x_{i}\right)$ the binary mask at the location $x_{i}$ that set to one if the data are sampled at an associated location and set to zero otherwise and $w_{x_{p}}$ is the normalizing factor, which is the sum of kernel weights. The adaptive steering kernel $K_{H_{i}{}^{steer}}$ is computed according to Equation (25) and the covariance matrix $C_{x_{p}}$ according to Equation (26).

## 5. Experimental Results

In our experiments, we used 26 images from the 32 (the others being resemblances) of the cave dataset [18], in which multispectral images consist of 31-band multispectral images acquired under illuminant D65. The 31-band images were acquired every 10 nm at between 400 and 700 nm. The image size was 512 × 512 pixels. The CAVE dataset is often used as a standard multispectral image dataset.

To evaluate the performance of the proposed algorithm, we compared it with recent four-band multispectral demosaicking methods, namely generic binary tree edge sensing (BTES) [9], directional filtering and wavelet transformation (DFWF) [6], adaptive spectral-correlation based demosaicking (ASCD) [4] and neighborhood in linear minimum mean square error (N-LMMSE) [16]. ASCD and N-LMMSE are a five-band and eight-band methods, respectively, which we implemented to four-band for comparison purposes. Visual and objective evaluations were also conducted.

### 5.1. Visual Performance Evaluations

For evaluation purposes, we selected four images with detailed structures as shown in Figure 11, Figure 12, Figure 13 and Figure 14. From the partially zoomed-in view images (red areas in original images), one notes the visible presence of blurring and false colors artifacts in the images demosaicked with the algorithms BTES, DFWF, ASCD, and N-LMMSE as is the case in Figure 12b–e, Figure 13b–e and Figure 14b–e which, respectively, display the green, blue, and orange bands of the feathers, hairs, and cloth images. In Figure 11b–e showing the red band of the face image, these artifacts are more visible with the BETS and N-LMMSE algorithms. In Figure 12, we note the presence of ghost noise in part of the reconstructed images with the BTES, DFWF, ASCD, and N-LMMSE algorithms. The quality of the reconstructed image is considerably reduced by these artifacts which are due to the lack of edge-preserving of the BTES, DFWF, ASCD, and N-LMMSE algorithms. Our proposed method reconstructs images without significant blurring or zipper artifacts (Figure 11f, Figure 12f, Figure 13f and Figure 14f). The four reconstructed images with our proposed demosaicking algorithm preserve details at edges and in textured areas better than the other algorithms. Overall, by comparing the results of the visual assessment, we can confidently say that our proposed method is better than the BTES, DFWF, ASCD, and N-LMMSE algorithms.

### 5.2. Quantitative Performance Evaluations

To quantitatively assess the objective performance of our proposed algorithm, we used the PSNR, SSIM, and RMSE metrics as described in [6,22] and calculated from the original and demosaicked images. The average values of PSNR, SSIM, and RMSE obtained from various algorithms are shown in Table 2, Table 3 and Table 4, respectively, such that the best scores are in bold. Note that the lower the value of RMSE, the better the performance of the algorithm.

A careful analysis of Table 2 shows that the DFWF algorithm produced the highest PSNR value which is 46.9371 while it is 42.3519 for the proposed method. Moreover, this method gives better PSNR scores for twelve images out of twenty-six while ours produces ten images. The N-LMMSE method gives no good score but the other two algorithms for two images each. Images for which the PSNR values are high for the DFWF algorithm are smoother images, but not textured images. The results of the average values of the PSNR show that our proposed method ranks first in the competition with DFWF, while N-LMMSE comes in last. In Table 3, according to the SSIM values, our algorithm outperforms all others with a higher value of the metric for both sixteen out of twenty-six images and for the average value. It is followed by BTES with a better score of the SSIM for ten images, the other three methods come in last. In Table 4, our algorithm produced better RMSE scores for seventeen images, unlike the others which have low scores. The RMSE values confirm the previous results with a lower mean value for our proposed method than the other algorithms. We can therefore see that overall, the values of PSNR, SSIM, and RMSE obtained with our algorithm are better than those of the BTES, DFWF, ASCD, and N-LMMSE methods.

## 6. Conclusions

In this study, we identified a four-band MSFA pattern for single-sensor cameras, arranged in a 6 × 6 moxel half filled with the blue band taking into account the properties of the liquid crystal tunable filter with which the sensor surface of the camera used is covered to acquire simulation images of the cave dataset. Based on the existing one, we then proposed a consequent algorithm that combines the LMMSE method and the adaptive kernel regression. In the proposed algorithm, we estimated the missing blue bands by the LMMSE method and the other spectral bands by the directional gradient method which relies on the estimated blue bands. Finally, applying the adaptive kernel regression gradient method to each spectral band refines the band by ridding it of artifacts that can adversely affect the reconstruction performance. In the experiment, we evaluated the proposed algorithm both visually and quantitatively with the existing algorithms BTES, DFWF, ASCD, and N-LMMSE. The results show that our proposed algorithm outperforms the others both visually and in terms of PSNR, SSIM, and RMSE.

The future work consists in deepening the algorithm in terms of the number of spectral bands by varying the moxels such as they are of 4 × 4 and 8 × 8 support to fully appreciate the degradation behavior of each spectral band.

## Figures and Tables

**Figure 1 jimaging-08-00295-f001:**
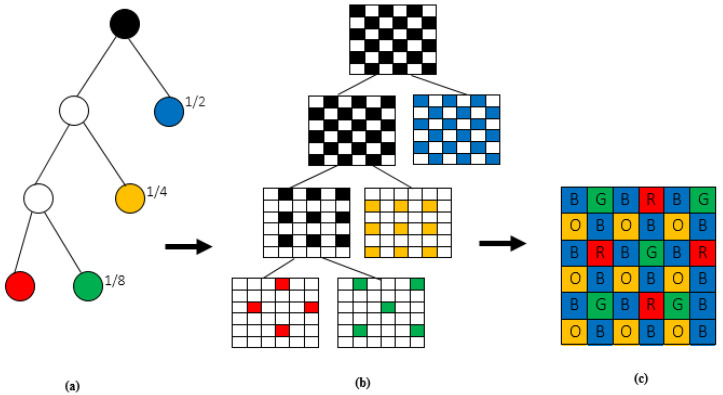
Four-band MSFA configuration: (**a**) binary tree considering appearance probabilities (**b**) decomposition and subsampling processes (**c**) MSFA configuration.

**Figure 2 jimaging-08-00295-f002:**
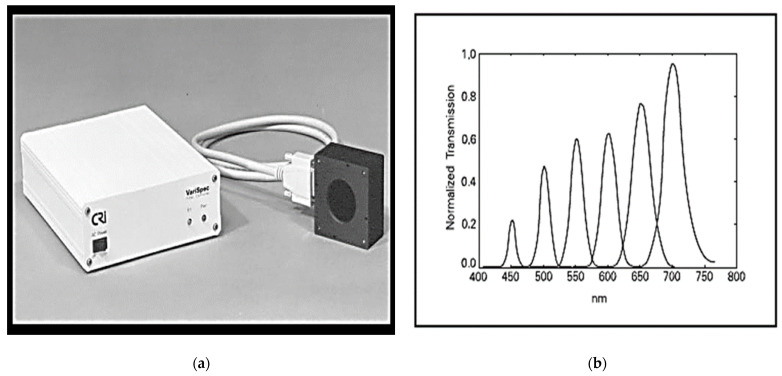
(**a**) The LCTF (**b**) The LCTF at several wavelength settings [19].

**Figure 3 jimaging-08-00295-f003:**
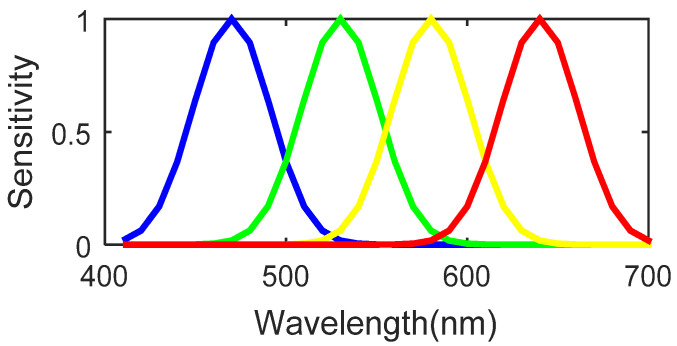
Spectral sensitivity of the 4-band filters.

**Figure 4 jimaging-08-00295-f004:**
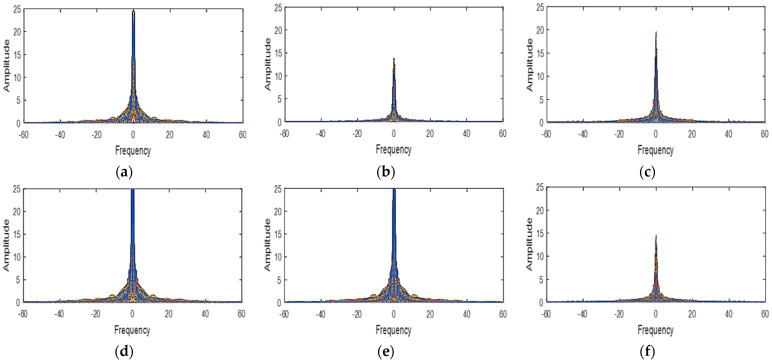
The power spectrum functions of the gradient signal in balloons image (**a**) green−red (**b**) green−blue (**c**) green–orange (**d**) red−blue (**e**) red−orange (**f**) blue−orange.

**Figure 5 jimaging-08-00295-f005:**
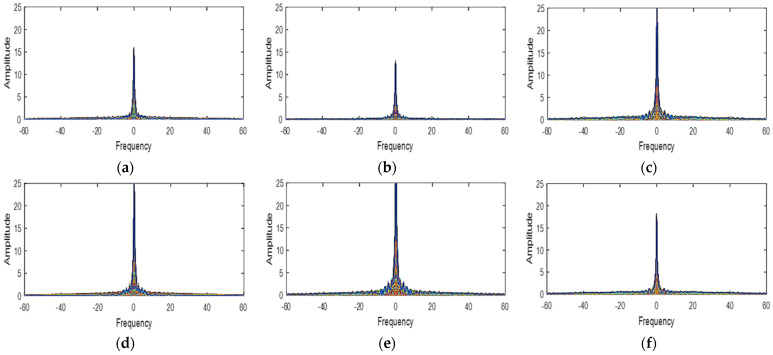
The power spectrum functions of the gradient signal in hairs image (**a**) green−red (**b**) green−blue (**c**) green−orange (**d**) red−blue (**e**) red−orange (**f**) blue−orange.

**Figure 6 jimaging-08-00295-f006:**
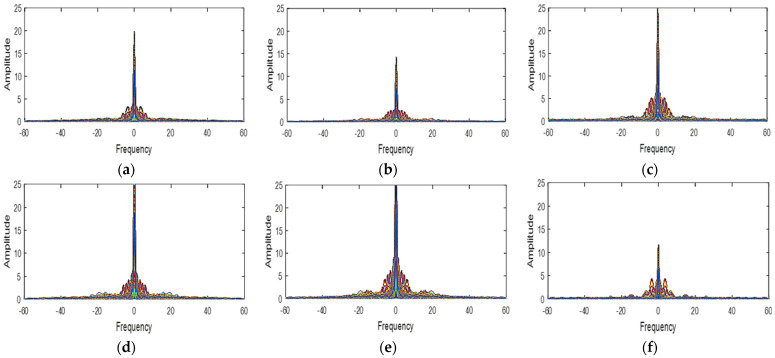
The power spectrum functions of the gradient signal in beers image (**a**) green−red (**b**) green−blue (**c**) green−orange (**d**) red−blue (**e**) red−orange (**f**) blue−orange.

**Figure 7 jimaging-08-00295-f007:**
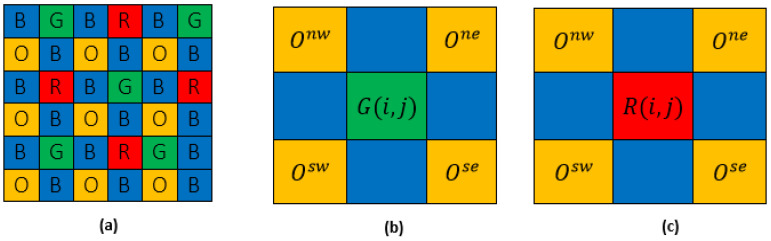
(**a**) Four-band MSFA (**b**) neighborhood of G-band (**c**) neighborhood of R-band.

**Figure 8 jimaging-08-00295-f008:**
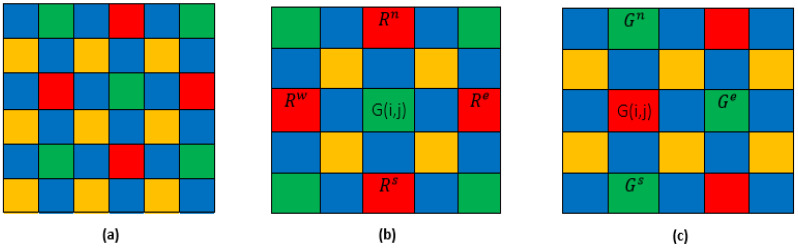
(**a**) Four-band MSFA (**b**) neighborhood of G-band (**c**) neighborhood of R-band.

**Figure 9 jimaging-08-00295-f009:**
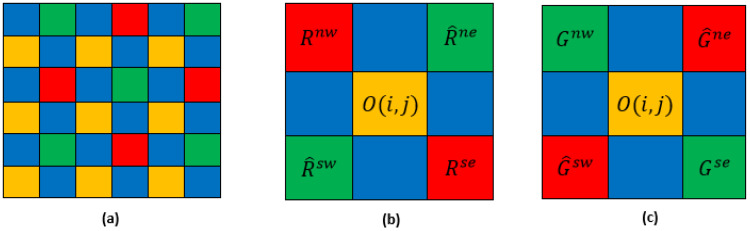
(**a**) Four-band MSFA (**b**) neighborhood of O-band for original and estimated red pixels (**c**) neighborhood of O-band for original and estimated green pixels.

**Figure 10 jimaging-08-00295-f010:**
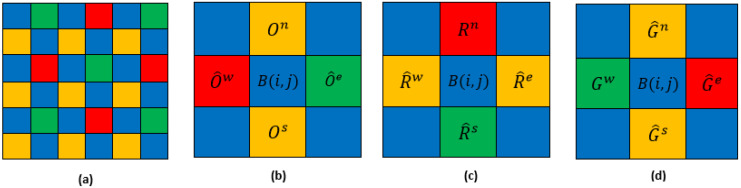
(**a**) Four-band MSFA (**b**–**d**) neighborhood of B-band for original and estimated orange, red and green pixels.

**Figure 11 jimaging-08-00295-f011:**
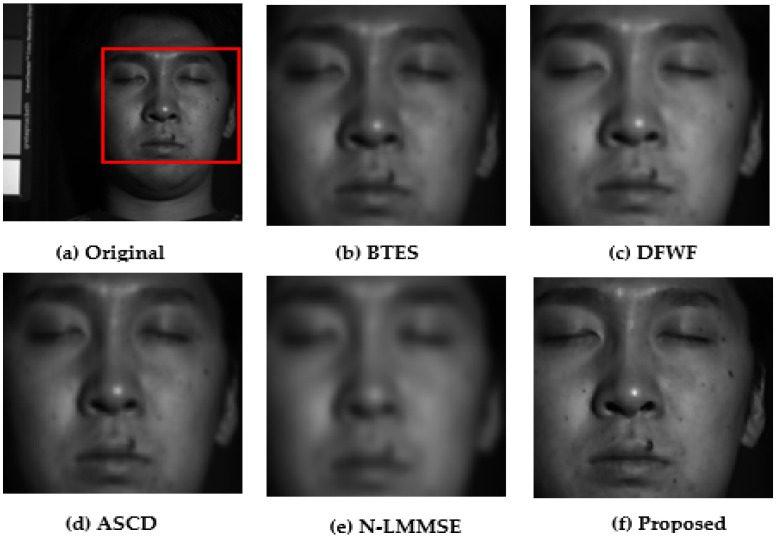
Visual comparison of red band in face image.

**Figure 12 jimaging-08-00295-f012:**
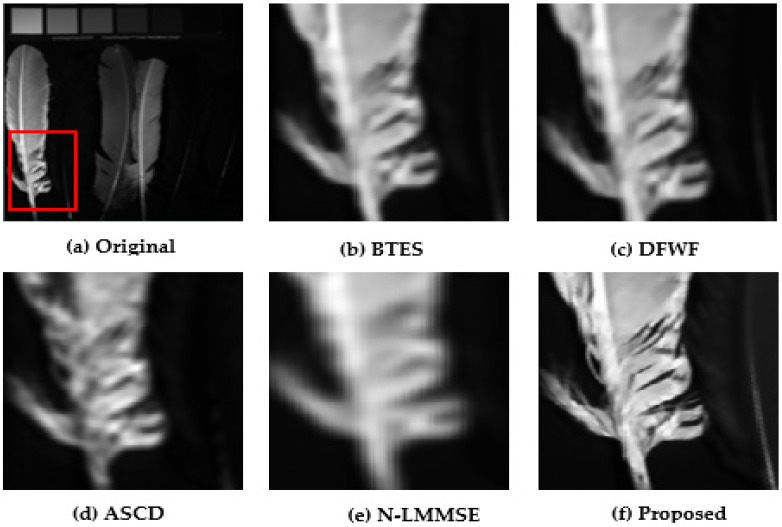
Visual comparison of red band in feathers image.

**Figure 13 jimaging-08-00295-f013:**
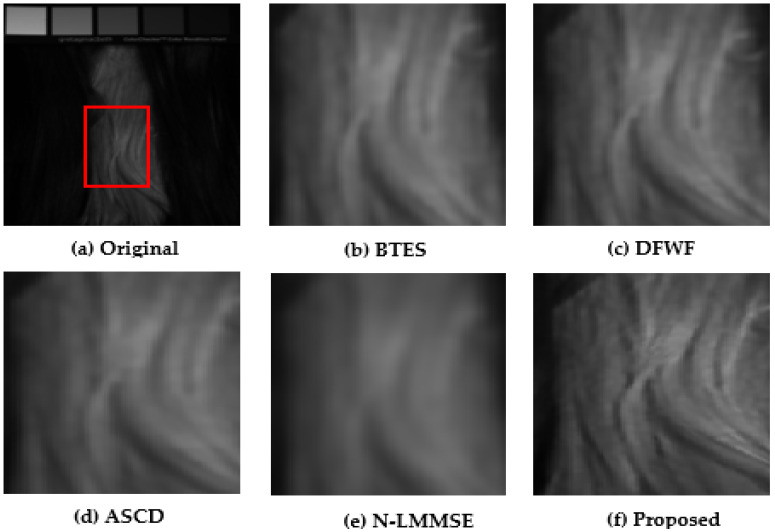
Visual comparison of red band in hairs image.

**Figure 14 jimaging-08-00295-f014:**
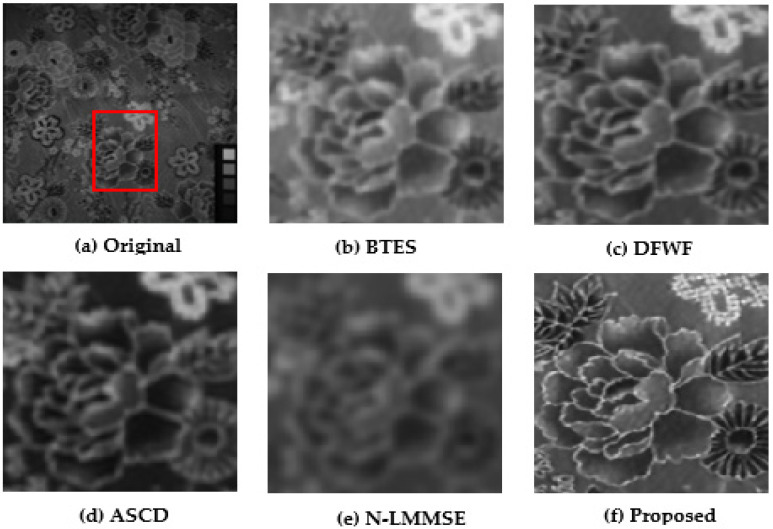
Visual comparison of red band in cloth image.

**Table 1 jimaging-08-00295-t001:** Spectral correlation coefficients for red/green ($C_{rg}$), red/blue ($C_{rb}$), red/orange ($C_{ro}$), blue/green ($C_{bg}$), blue/orange ($C_{bo}$) and orange/green ($C_{og}$).

Images	$$C_{rg}$$	$$C_{rb}$$	$$C_{ro}$$	$$C_{bg}$$	$$C_{bo}$$	$$C_{og}$$
Beads	0.5469	0.4267	0.2773	0.6887	0.8381	0.2498
Balloons	0.8000	0.6647	0.6529	0.9602	0.9666	0.8968
Pompoms	0.6138	0.1869	0.0152	0.7798	0.9021	0.5075
Cloth	0.9700	0.9222	0.6599	0.9654	0.8180	0.6679
Statue	0.9818	0.9413	0.8688	0.9859	0.9797	0.9356
Face	0.9817	0.9530	0.8665	0.9896	0.9637	0.9248
Food	0.9882	0.9008	0.6753	0.9362	0.9168	0.7244
Feathers	0.8995	0.8398	0.7235	0.9733	0.9046	0.8378
Flowers	0.9247	0.7965	0.6781	0.9438	0.9317	0.7952
Beans	0.9454	0.9136	0.8465	0.9652	0.9576	0.8772
Painting	0.9610	0.8192	0.6973	0.9348	0.9674	0.8332
Thread	0.9002	0.8467	0.7339	0.9518	0.9326	0.7982
Clay	0.6538	0.5807	0.2737	0.7703	0.7579	0.200
Superballs	0.6448	0.7475	0.5906	0.7482	0.8802	0.3602
Toys	0.9685	0.8780	0.6065	0.9441	0.8567	0.6685
Glass	0.7431	0.4076	0.2074	0.8667	0.8476	0.5632
CD	0.828	0.7183	0.7070	0.8780	0.7761	0.5789
Hairs	0.9814	0.9524	0.8965	0.9919	0.9862	0.9591
Peppers	0.9049	0.7097	0.5233	0.9265	0.8626	0.6935
Sponges	0.5476	0.3080	0.0675	0.9068	0.8371	0.5745
Paints	0.9744	0.9525	0.9001	0.9892	0.9583	0.9219
Beers	0.9461	0.8127	0.6932	0.9524	0.9772	0.8751
Chart_Toy	0.9951	0.9866	0.9674	0.9964	0.9906	0.9792
Sushi	0.9813	0.9559	0.7866	0.9804	0.8892	0.7964
Lemons	0.8715	0.7262	0.6711	0.9658	0.9897	0.9325
Slices	0.9443	0.8963	0.8522	0.9838	0.9679	0.9287

**Table 2 jimaging-08-00295-t002:** The PSNR average results of demosaicking algorithms.

Images		PSNR			
	BTES	DFWF	ASCD	N-LMMSE	Ours
Beads	30.7458	**33.2131**	30.2940	28.1932	29.8430
Balloons	42.0289	**46.9371**	39.2510	38.5023	40.0571
Pompoms	38.4598	**41.2875**	35.1001	31.7721	33.7244
Cloth	28.5308	31.3640	28.9022	33.1767	**34.1697**
Statue	40.6305	**44.1420**	31.5929	39.8950	41.2060
Face	38.2092	40.2888	35.9277	38.1173	**41.1619**
Food	40.0772	**43.2572**	37.0315	40.0100	40.0189
Feathers	35.1460	**39.4372**	33.0949	31.2218	34.8913
Flowers	**39.1085**	38.4263	33.0538	33.6341	38.6609
Beans	32.6284	**36.9307**	32.2185	29.4240	34.0663
Painting	30.8851	**34.8590**	28.4571	30.9910	34.7533
Thread	36.3351	**41.3007**	31.6812	35.3227	39.5297
Clay	32.2509	**36.1485**	34.3987	31.2567	34.4575
Superballs	41.7985	**44.9294**	36.3779	34.7720	37.0399
Toys	42.7080	**43.4266**	36.7039	35.6316	38.8146
Glass	26.4927	31.1506	31.3545	30.8802	**33.5763**
CD	36.4992	34.8518	37.8778	36.2179	**39.8332**
Hairs	32.9339	36.8394	36.2247	36.8732	**39.9698**
Peppers	35.0235	33.4378	36.0790	34.4836	**36.8838**
Sponges	30.5707	25.5476	**31.2702**	29.3916	30.1159
Paints	27.2903	28.2379	32.4600	33.0013	**33.6658**
Beers	**36.1153**	29.1305	33.6159	30.5370	33.5116
Chart_Toy	28.0560	31.1273	32.6595	34.7302	**37.9766**
Sushi	37.2125	38.8466	39.4100	40.0039	**42.3519**
Lemons	31.9442	35.3157	**38.6506**	32.8326	36.7108
Slices	31.0185	35.2776	35.3538	38.1607	**40.1883**
Average	34.7192	36.7581	34.1939	34.1935	**36.8146**

**Table 3 jimaging-08-00295-t003:** The SSIM average results of demosaicking algorithms.

Images		SSIM			
	BTES	DFWF	ASCD	N-LMMSE	Ours
Beads	0.8719	0.7840	0.8368	0.8073	**0.8756**
Balloons	**0.9903**	0.9398	0.9449	0.9561	0.9748
Pompoms	**0.9549**	0.8606	0.8956	0.9001	0.9046
Cloth	0.8476	0.9155	0.7864	0.9153	**0.9278**
Statue	0.9413	0.9739	0.9354	0.9419	**0.9797**
Face	0.9718	0.9830	0.9471	0.9528	**0.9881**
Food	0.9749	0.9667	0.9649	0.9675	**0.9804**
Feathers	0.9480	0.9197	0.8924	0.9208	**0.9482**
Flowers	0.9519	0.9165	0.8917	0.9394	**0.9575**
Beans	**0.9524**	0.9019	0.8836	0.9190	0.9356
Painting	0.8798	0.8887	0.7743	0.8985	**0.9127**
Thread	0.9208	0.9561	0.8757	0.9577	**0.9731**
Clay	**0.9780**	0.9017	0.8837	0.9216	0.9455
Superballs	**0.9807**	0.9156	0.9299	0.9399	0.9548
Toys	0.9701	0.9513	0.9269	0.9522	**0.9769**
Glass	0.9145	0.8843	0.8681	0.9148	**0.9392**
CD	**0.9791**	0.9378	0.9470	0.9560	0.9741
Hairs	0.9536	0.9717	0.9127	0.9654	**0.9785**
Peppers	**0.9842**	0.9058	0.8879	0.9132	0.9478
Sponges	**0.9693**	0.8927	0.8862	0.8890	0.9014
Paints	0.9380	0.9159	0.9195	0.9300	**0.9690**
Beers	**0.9766**	0.9490	0.9026	0.9565	0.9705
Chart_Toy	0.9338	0.9538	0.9131	0.9696	**0.9740**
Sushi	0.9749	0.9443	0.9726	0.9698	**0.9817**
Lemons	**0.9609**	0.9353	0.9518	0.9355	0.9578
Slices	0.9455	0.9278	0.9329	0.9549	**0.9751**
Average	0.9486	0.9228	0.9025	0.9325	**0.9542**

**Table 4 jimaging-08-00295-t004:** The RMSE average results of demosaicking algorithms.

Images		RMSE			
	BTES	DFWF	ASCD	N-LMMSE	Ours
Beads	0.0407	0.0398	0.0347	0.0470	**0.0340**
Balloons	0.0164	0.0160	0.0162	0.0191	**0.0140**
Pompoms	0.0279	0.0273	**0.0237**	0.0288	0.0257
Cloth	0.0342	0.0315	0.0408	0.0201	**0.0197**
Statue	0.0121	0.0118	0.0262	0.0113	**0.0096**
Face	0.0118	0.0114	0.0178	0.0169	**0.0100**
Food	0.0137	0.0134	0.0173	0.0171	**0.0109**
Feathers	0.0227	0.0218	0.0286	0.0229	**0.0184**
Flowers	0.0176	0.0189	0.0247	0.0175	**0.0139**
Beans	0.0245	0.0232	0.0279	0.0255	**0.0209**
Painting	0.0207	**0.0163**	0.0396	0.0208	0.0189
Thread	0.0199	0.0186	0.0303	**0**.0182	**0.0113**
Clay	**0.0101**	0.0315	0.0320	0.0290	0.0208
Superballs	0.0217	0.0212	0.0205	0.0168	**0.0152**
Toys	0.0181	0.0180	0.0239	0.0157	**0.0132**
Glass	0.0228	0.0322	0.0314	0.0236	**0.0218**
CD	**0.0076**	0.0202	0.0168	0.0127	0.0115
Hairs	0.0117	0.0156	0.0183	0.0140	**0.0103**
Peppers	**0.0087**	0.0244	0.0236	0.0163	0.0152
Sponges	**0.0172**	0.0679	0.0445	0.0441	0.0401
Paints	**0.0210**	0.0395	0.0287	0.0280	0.0216
Beers	**0.0120**	0.0363	0.0260	0.0277	0.0225
Chart_Toy	0.0209	0.0277	0.0269	0.0153	**0.0129**
Sushi	0.0107	0.0127	0.0123	0.0108	**0.0082**
Lemons	**0.0142**	0.0207	0.0144	0.0176	0.0155
Slices	0.0135	0.0175	0.0190	0.0126	**0.0100**
Average	0.0179	0.0244	0.0256	0.0211	**0.0171**

## Data Availability

The multispectral images used for our simulations are images from the CAVE dataset and can be downloaded from: http://www.cs.columbia.edu/CAVE/databases/multispectral/ (accessed on 16 September 2022).

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
