# Peer review of "4-Band Multispectral Images Demosaicking Combining LMMSE and Adaptive Kernel Regression Methods"

_2313-433X, 2022, doi:10.3390/jimaging8110295_

Round 1

Reviewer 1 Report

As attached.

Author Response

Hello.

We thank you so much for your interest for our work.

Reviewer 2 Report

Publish with a few corrections as listed in enclosed document

Author Response

(The authors gave the same response as above.)

Reviewer 3 Report

Accept in the present form.

Author Response

(The authors gave the same response as above.)

Reviewer 4 Report

In this work 4-band multispectral images demosaicking combining LMMSE and adaptive kernel regression methods was proposed. The analysis carried out by the authors is extremely interesting. However the work would appear to be in some of the section  poor, badly described and missing in some of its parts and therefore it cannot be published at present. A great deal of effort must be put into the organisation of materials, methods and results. In addition, the entire discussion section needs to be revised.

Abstract: Please add some preliminary result at the end of this section.

Introduction: the authors provide a good literature overview about the present work, however they should highlight more what they work add and the authors should pay more attention to elaborate on the novelty and contribution to the state of the research through their work. From the current state of the manuscript, this

Discussions: This section needs to be completely revised. There is no comparison with the literature. The results obtained must be discussed and compared with other works.

Author Response

(The authors gave the same response as above.)

Round 2

Reviewer 1 Report

The energy balance issue has been clarified. The improvement using the proposed method is more clearly demonstrated.

The captions of Figures 5(a)-(f) and Figures 6(a)-(f) should be given.

Author Response

Thank you for your interest in our work. We have provided answers to your concerns through the following line.

Point 1 : The captions of Figures 5(a)-(f) and Figures 6(a)-(f) should be given.

Response 1 : The captions of Figures 5 (a)-(f) and Figures 6 (a)-(f) are given.

Reviewer 4 Report

I recognise the work of the authors to improve the work. Unfortunately, however, I do not see significant differences in the discussion section... In this section the results are not compared with existing work and discussed on the basis of the differences that emerged to highlight the strengths of the proposed work... In my opinion this part is crucial and consequently the work for me still needs to be revised extensively in this part... 

Author Response

Thank you for your interest in our work. We have provided answers to your concerns through the following lines.

Point 1: I recognise the work of the authors to improve the work. Unfortunately, however, I do not see significant differences in the discussion section... In this section the results are not compared with existing work and discussed on the basis of the differences that emerged to highlight the strengths of the proposed work... In my opinion this part is crucial and consequently the work for me still needs to be revised extensively in this part... 

Response 1 : We have tried in our own way to improve the discussion on the comparison of the different methods with our proposal in the last paragraph before the conclusion.